# Identifying Token-Level Dialectal Features in Social Media

**Jeremy Barnes**[1], **Samia Touileb**[2], **Petter Mæhlum**[3], **Pierre Lison**[4]

[1]University of the Basque Country, [2]University of Bergen,
[3]University of Oslo, [4]Norwegian Computing Center
`jeremy.barnes@ehu.eus, samia.touileb@uib.no,`
`pettemae@ifi.uio.no, plison@nr.no`

## Abstract

Dialectal variation is present in many human languages and is attracting a growing interest in NLP. Most previous work concentrated on either classifying dialectal varieties at the document or sentence level or performing standard NLP tasks on dialectal data. In this paper, we propose the novel task of *token-level dialectal feature prediction*. We present a set of fine-grained annotation guidelines for Norwegian dialects, expand a corpus of dialectal tweets, and manually annotate them using the introduced guidelines. Furthermore, to evaluate the learnability of our task, we conduct labelling experiments using a collection of baselines, weakly supervised and supervised sequence labelling models. The obtained results show that, despite the difficulty of the task and the scarcity of training data, many dialectal features can be predicted with reasonably high accuracy.

## 1 Introduction

Language variation is a pervasive phenomenon in human language. These varieties can differ on phonemic, lexical, or syntactic levels, among others, and often vary on several levels at a time (Chambers and Trudgill, 1998). One common type of language variation stems from geographical location, as people actively use regional variations to mark their identity. When a language variety indicates *where* a speaker is from, we call this variety a **dialect**, or more precisely a **geolect** or **topolect**, as the word 'dialect' can also refer to social background or occupation. In this work, we use 'dialect' to denote geographical variation.

Dialectal variation in Norwegian is widespread and, in contrast to many languages, the use of spoken and written dialects in the public sphere is generally viewed positively (Bull et al., 2018). Although Norwegian can be broadly divided into four dialectal regions, many dialectal features are shared across these regions (see Figure 1). Therefore, rather than seeing dialects as discrete categories, we should view them as a combination of correlated dialectal features (Nerbonne, 2009).

The under-resourced status of dialects, however, makes it difficult to build NLP tools from scratch. This is exacerbated by the growing reliance on pre-trained language models, which often encounter few examples of dialectal data during training. If NLP models fail to process dialectal inputs, their deployment may reinforce existing inequalities, as those who use a non-standard variety will either receive worse service or be forced to adopt a standard variety to interact. Those who advocate for maintaining dialectal variation also depend on tools to help them monitor the use of dialects on social media. This motivates the development of fine-grained models of dialectal features.

Previous work on dialectal NLP has classified dialects, geographical location, or provided training and testing resources for various dialects. In this paper, we take a different viewpoint on identifying dialects, opting to label the *token-level dialectal features* of a text rather than classifying or predicting the geolocation of the entire text. We first propose a fine-grained annotation scheme for token-level dialectal features in Norwegian. We then annotate a corpus of Norwegian dialectal tweets using this scheme, and finally validate its use for fine-tuning neural sequence labeling models in Norwegian.

Our contributions are 1) we introduce the novel task of **token-level dialect feature identification**, 2) provide a **novel corpus of Norwegian dialectal tweets** annotated for 21 token-level features,[1] and 3) describe **extensive experiments** demonstrating the learnability and difficulty of the task.

---

[1]Annotation guidelines, procedure and data available at
`https://github.com/jerbarnes/nordial`

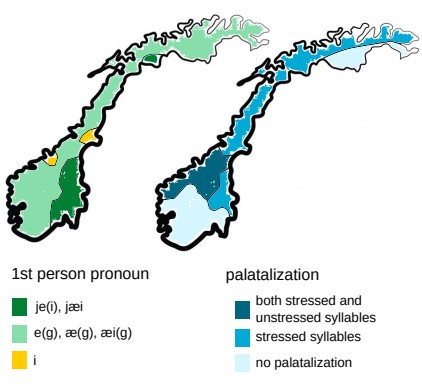

1st person pronoun
- je(i), jæi
- e(g), æ(g), æi(g)
- i

palatalization
- both stressed and unstressed syllables
- stressed syllables
- no palatalization

Figure 1: Map of two dialectal features in Norwegian that do not coincide geographically.

## 2 Related Work

In contrast to more formal writing, social media abounds with dialectal variation, ranging from variation between racial groups (Eisenstein, 2015), to variation within online communities (Danescu-Niculescu-Mizil et al., 2013). While not all levels of variation are equally present, often due to a speaker's lack of awareness of sociolinguistic indicators (Labov, 2006), a substantial share of dialectal variation is reliably transcribed in social media posts (Eisenstein, 2013; Doyle, 2014).

For NLP, dialectal data presents both a challenge and an important area to improve upon. Previous work in NLP has included descriptive corpus studies (Jones, 2015; Tatman, 2016), dialect classification (Zampieri et al., 2017), geolocation of tweets based on their dialectal features (Eisenstein et al., 2010; Hovy and Purschke, 2018) or quantifying the spatial dependence of linguistic variables (Nguyen and Eisenstein, 2017).

There have also been a series of workshops (VarDial) (e.g. Nakov et al., 2016, 2017; Zampieri et al., 2018) that include work on discriminating similar languages (Haas and Derczynski, 2021), identifying dialects (Jauhiainen et al., 2021), and geolocation of tweets (Gamăn et al., 2021). The workshops have also held several shared tasks with the aim to identify languages and dialects (Zampieri et al., 2017), as well as morpho-syntactic tagging (Zampieri et al., 2018). Another series of shared tasks have focused on the identification of Arabic dialects (Bouamor et al., 2019; Abdul-Mageed et al., 2020, 2021). While each of these shared tasks proposed dialect identifications on different level of granularity (region, country, and city-levels),

they all approached dialect identification as a sentence classification task. Work on code-switching (e.g. Solorio and Liu, 2008; Jain and Bhat, 2014; Samih et al., 2016; Çetinoğlu, 2016), on the other hand, has focused on word-level classification, but usually casts this a binary decision, rather than identifying fine-grained labels.

Regarding Norwegian dialects specifically, linguistic work is long and varied. Christiansen (1954) described the main dialect regions, while Sandøy (2000) describes several factors that drive language change in modern Norwegian dialects, e.g. urban jumping (Chambers and Trudgill, 1998), the prestige of certain dialects, or the general tendency towards simplification. Within NLP, Barnes et al. (2021) present the *NorDial* corpus, a curated collection of 1,073 tweets classified as either Bokmål, Nynorsk (which are the two standardized written forms for Norwegian), dialectal, or mixed. The authors experimented with classifying these tweets with Norwegian BERT models (Kummervold et al., 2021) and found that the resulting models achieved reasonably good performance at identifying tweets written in dialectal Norwegian.

Demszky et al. (2021) introduce the task of dialect feature detection at the phrase/sentence level. They use available annotations on the ICE-India English data (Greenbaum and Nelson, 1996) and annotate a small amount of this data with separate set of 18 dialectal features. As they have no training data for their annotated features, they propose to use a minimal pairs framework as a kind of weak supervision. They find that even with minimal supervision, their models are able to reliably predict many of the features. However, they do not predict which tokens carry the features, choosing to label the entire phrase instead.

To address these limitations, we propose a new approach where we annotate dialectal features at the token level. We contend that this annotation strategy provides a more fine-grained view of the actual use of dialectal features in social media.

## 3 Dialectal tweet collection

In order to increase the number of dialectal tweets, we expand upon the NorDial corpus (Barnes et al., 2021) and collect a further 3,000 tweets to annotate. During the initial collection, we used the Twitter API without a search query and confined the search to tweets from the geographical area of Norway. This first collection, however, yielded relatively

```
     ...    y'all   fixin'    to    leave?
            subj-pron  lexical        lexical
                       g-drop

         'are you-pl about to leave?'
```

Figure 2: Example of Texan English with dialectal labels below each token.

few dialectal tweets and those found displayed a narrow set of dialectal features. To increase the variety of dialectal features, we first collected a list of dialectal features from the *Store Norske Leksikon*[2] (Norwegian Encyclopedia) and used these as queries in the Twitter API. We then identified users whose tweets often contain these dialectal features and collected their tweets, as well as tweets from their followers. As many of the collected tweets were still written in standard Bokmål or Nynorsk, three annotators were asked to classify the tweets, and those labelled as dialectal were then included in the process of fine-grained feature annotation. In total, 2,455 of 3000 tweets were classified as dialectal.

## 4 Annotation of fine-grained dialectal features

Figure 2 shows an example from Texan English with three main dialectal features: *y'all*, which is the non-standard second person plural pronoun and *fixin' to*, which contains the lexical feature 'fixing to' which means 'about to', and the morphological feature of 'g-dropping'.

In the rest of this section, we detail the inventory of dialectal features used in our annotation. As each example highlights a minimal pair example of a single dialectal feature, we do not include the labels below the relevant tokens.

### 4.1 Dialectal features

The inventory of dialectal features stems from the linguistic traits that can be encountered in written form as described by Venås and Skjekkeland (2022). Other dialectal features, such as differing toneme patterns or the pronunciation of 'l', were not considered, as they are not observable in written texts. We focus on the dialectal impact a word

---

[2]https://snl.no/

has, *i.e.* whether the annotator can determine that the word falls outside of the norms in such a way as to identify the speaker as a dialect user. For example, a form like *jæ* for 'I' has a higher impact than the choice between the two habitual aspect markers *bruke* and *pleie*, 'use (to)', as the latter are both part of the written norm, and the former is unlikely to be an accidental misspelling.

In cases where there are several choices of form, some of these might be more marked than others. In the following examples, we show the original dialectal version and normative Bokmål versions: **dialect**/**normative** and the English translation.

**Subject and object pronoun use**    Pronouns are extremely common dialect markers in Norwegian, as a single pronoun can be marked enough to identify the dialect of the writer. We label the subject and object (or oblique) functions separately, but do not include a separate label for the dative.

(1)   ... og **dem**/**de** blir aldrig eldre ...
      '... and they never get older ...'

**Copula**    The copula 'være/vera/vere' (be) is marked with the label copula. We only mark dialectally interesting, non-standard versions of the copula, such as 'e' and 'værra'.

(2)   Det **e**/**er** rart at ...
      'It is weird that ...'

**Contraction**    We label contractions for negation adverb 'ikke/ikkje' (not), and enclitic pronouns. The verb and the adverb are labeled separately, but both are labeled with the *contraction* label.

(3)   **ekke**/**er ikke** han som skulle ...
      'he is not the one who should have' ...

**Palatalization**    In Norwegian palatalization occurs frequently to geminated consonants such as 'nn', 'dd' and 'll', in several dialects. In writing it is usually indicated by additions of 'j' or 'i'.

(4)   ho e nok **forbainna**/**forbanna** ...
      'She is so angry ...

**Present marker deletion**    In some dialects the final '-r' that marks the present tense for many verbs in both Bokmål and Nynorsk is dropped. We also use this label to indicate the dropping of '-l' in present tense verb forms such as 'skal' → 'ska' (will) and 'vil' → 'vi' (want).

**Apocope**    Apocope is the loss of word-final '-a' or '-e' and is common in certain dialects.

(5)    Æ e her for å **vinn**/**vinne**

    'I am here to win' ...

**Voicing**    Voicing is the process by which consonants which are voiceless in some dialects become voiced, where 'p', 't', and 'k' become 'b', 'd', and 'g', respectively.

(6)    Eg kommer ikkje **tebage**/**tilbake**

    'I won't come back' ...

**Vowel shift**    Both monophtongal changes such as lowering (e→æ) and dipthongization such as 'e' → 'ei' are all marked with the vowel shift label. We also see cases of monophthongization such as 'ei' → 'ø'. One important heuristic we follow is that we do not mark vowel shift in words that are tagged with any of the pronoun labels.

**Lexical variation**    This label is used when the lemma of a word is notably marked. Loanwords are not affected by this; the word has to be a dialectal or local version of a standard word that could have been used instead. An example is the word 'tue' (towel) instead of 'klut' (cloth).

**Demonstrative pronoun use**    In some dialects it is common to use third-person pronouns as determiners together with proper names. These can be full forms as in 'ho Kari' (she Kari) or 'han Olav' (he Olav) or reduced as in 'a Kari' or 'n Olav'.

**Shortening**    In some dialects, writers indicate a change of stress to the first syllable with accompanying vowel reduction and consonant lengthening, by writing a double consonant after the first syllable if there is originally only one, as in 'pottet' instead of 'potet' (potato).

**Grammatical gender of nouns**    The grammatical gender of nouns in Norwegian has considerable variation. The least common remnant of the feminine gender is the indefinite article 'ei'. Keeping the feminine definite form '-a' is more common, but there is also a clear tendency to see certain high-frequency words as feminine. Examples are words like 'jente' (girl). 'Ei jente' (a girl) is slightly marked towards favoring the feminine form, while 'jenten' (the girl) is strongly marked towards a dialect with no feminine gender.

**Marked**    This label is used for words that are part of the written languages' norms, but which are still rarely used, and therefore dialectally marked. An example is the question word 'åssen' (how), which is accepted in Bokmål, but still infrequent, and somewhat marked compared to 'hvordan' (how).

**h-v**    A notable difference between Bokmål and Nynorsk is that Nynorsk has 'kv' where bokmål has 'hv', especially for interrogatives. In some dialects, the 'v' is lost, giving only 'k' or 'h', as in *'hårr'* for 'hvor' (where) or *'ka'* for 'hva' (what). This is marked with the *h-v* label. Any token with this label will not have the *phonemic spelling* label.

**Adjectival declension**    This labels is used for adjectives with non-standard endings, such as '-e' in indefinite or non-plural environments.

(7)    ein **gode**/**god** venn

    'a good friend' ...

**Nominal declension**    This label is used when a noun takes a non-standard declensional ending.

(8)    Fortsatt gode **muligheta**/**muligheter** til gå

    'still good chances to go' ...

**Verb conjugation**    This label is used when a verb takes a non-standard conjugation ending, such as 'skrivi' for 'skrive' (to write).

**Functional words**    The dialectal forms of many functional words are spelled radically different. We label all functional words whose spellings are not in accordance with the written norms.

(9)    Tru **ittæ**/**ikke** dæ æ dær

    'do not think it is there' ...

**Phonemic spelling**    In cases where there is no clear dialectal variation, but it is clear that the speaker wants to indicate that they are writing a more oral form, the label phonemic spelling is used. This is mostly for cases where a pronunciation is close to the perceived norm of some standard, like 'næi' for 'nei' (no).

**Interjection**    This label is used for all interjections, dialectal or not, such as the greeting 'heia' (hey).

## 4.2 Annotation procedure

For the token-level annotations, we take the tweets that were classified as dialectal in the first round, combined with the dialectal tweets from Nordial (Barnes et al., 2021). The annotation was performed by three hired student research assistants with a background in linguistics and with Norwegian as native language. All annotators are from eastern Norway, and native speakers of the eastern dialect. The first 50 tweets were annotated independently by two annotators. This first round provided the basis for group discussions, held regularly during the first phase of annotation, after which the guidelines were updated. The doubly annotated documents were then adjudicated by a third annotator after a final round of discussions concerning difficult cases. Annotators had the possibility to discuss any potential problems during both the annotation and adjudication period, but were encouraged to follow the guidelines as strictly as possible. The annotation and adjudication were both performed using the web-based annotation tool Brat (Stenetorp et al., 2012).

## 4.3 Annotation results and statistics

Table 1 presents the statistics for the final annotated data. We create separate test and developments splits of 500 and 300 tweets respectively, maintaining the overall distribution of labels evenly throughout the splits and leave the remaining 1,655 tweets as training data. The average length of the tweets is around 25 tokens, with an average of 4.5 annotations per tweet. Most tokens in a tweet are not annotated (84.3%), leaving an average of 0.2 annotations per token. Of the remaining 15.7%, the average number of labels per token is 1.2. In other words, 14% (1343 tokens) of the annotated tokens have multiple labels, while the remaining 86% (8167 tokens) have a single label.

Figure 3 shows the distribution of the annotated labels. Vowel shift is the most common label, followed by subject pronoun, and functional. This is expected as vowel shift covers a large number of phenomena, and subject pronoun and functional are highly salient features in Norwegian dialects. The least common are interjection, demonstrative pronoun, and gender. While these features may be more common in spoken dialects, it seems writers of tweets use them less frequently, possibly because they are much more marked when written. See the Appendix for further analysis.

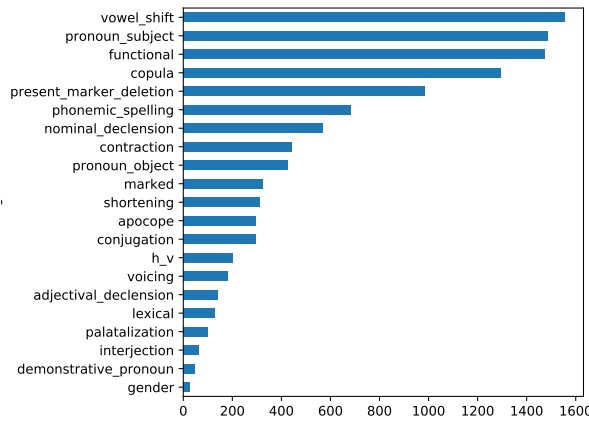

Figure 3: Frequency counts of dialectal features annotated in the full dataset of Norwegian tweets.

After completing the annotation process, the annotators pointed out that some dialectal areas (especially the Trøndersk-Central dialect) seem to be more common in the data. This might skew the label distribution to a degree.

## 4.4 Inter-annotator agreement

Chance-corrected inter-annotator agreement is important to determine the reliability of annotated data. The annotation we propose requires *unitization* or delimiting spans of words, *categorization*, and is inherently *multi-label*. Typical inter-annotator agreement measures, *e.g.* $\kappa$ (kappa) (Cohen, 1960) or $\alpha$ (alpha) (Krippendorff, 1980), do not provide a good statistical basis for determining agreement with multi-labels which can span several tokens. We therefore use the $\gamma$ (gamma) agreement from Mathet et al. (2015) instead, which allows for chance corrected agreement between annotators given the three above requirements.

$\gamma$ combines alignment and comparing of categorization into a single chance-corrected metric. It first selects the alignment that leads to the least overall disagreement $\gamma_o$ and then calculates the expected disagreement $\gamma_e$ by sampling from the existing annotations. Finally, as with other measures based on disagreement, gamma is calculated as $\gamma = 1 - \frac{\gamma_o}{\gamma_e}$, where the observed measure is divided by the expected measure. Values in gamma range from $-\infty$ to 1, where 0 represents chance agreement. We use the `pygamma-agreement` package (Titeux and Riad, 2021) available in python.

The double annotations from the first and second round achieve $\gamma = 0.63$, and $\gamma = 0.64$ respectively, which we take to indicate good agreement, given

| | train | dev | test | total |
|---|---|---|---|---|
| number of tweets | 1,655 | 300 | 500 | 2,455 |
| number of tokens | 40,483 | 7,563 | 12,597 | 60,643 |
| average number of tokens per tweet | 24.5 | 25.2 | 25.2 | 24.7 |
| average number of annotations per tweet | 4.5 | 4.4 | 4.5 | 4.5 |
| average number of annotations per token | 0.2 | 0.2 | 0.2 | 0.2 |
| average number of labels per annotated token | 1.2 | 1.2 | 1.2 | 1.2 |

Table 1: Statistics of the dialectal feature annotations.

that the task is challenging. Common disagreements between annotators include whether a token should be considered functional or not, the use of the lexical label and the identification of vowel shift.

## 5 Experiments

We now describe the experimental setup employed to validate our annotations. As early results indicated that standard models had difficulty learning multi-label sequence labelling tasks, we merge occurrences of multiple labels, yielding a total of 159 combinations (including 'Ø', the null label). For each possible combination of labels in our dataset, we create a new merged label that represents them. This increases the number of total classes to be predicted, but reduces the task to a much simplified multi-label sequence labelling problem. Formally, the task is then given a sequence of $N$ tokens $S = \{t_1, t_2, \ldots, t_n\}$ to predict the sequence of token-wise labels $L = \{l_1, l_2, \ldots, l_n\}$, where these labels can be either single labels, *e.g.*, 'vowel_shift' or a merged label, *e.g.*, 'lexical-vowel_shift'. For all experiments with neural models, we train an set of five models with different random seeds and report both micro-averaged $F_1$ and standard deviation.

### 5.1 Initial baseline

The first baseline consists of a simple majority voter that always predicts the most common label, which is 'vowel shift'.

### 5.2 Handcrafted functions

To investigate the extent to which the dialectal features can be inferred from known linguistic rules, we designed a set of handcrafted functions. One team member with a linguistics background and access to the annotation guidelines and the labelled training data implemented a set of 39 programmatic labeling functions, divided in three groups:

**Heuristic functions** Many labels can be detected programmatically. For example, to identify di-

alectal demonstrative pronouns, we create a function that detects demonstrative pronouns occurring within two tokens after a proper name.

**Lexicon functions:** Categories such as *h-v*, *functional*, or *interjection* correspond to (roughly) closed classes which can be directly compiled in lexicons. We also construct lexicons for other categories such as *marked* or *phonemic spelling*, although those categories are more productive and are not restricted to a closed set. Those lexicons are created by enumerating tokens associated with the corresponding tag in the development set.

**Dictionary-based functions** We can also predict a *voicing* tag by changing a soft consonant ('b', 'd', 'g') to its hard consonant ('p', 't', 'k') and then performing a lookup in precompiled dictionaries for Bokmål and Nynorsk[3].

The results of all labelling functions can then be aggregated into a unified prediction over possible labels. This aggregation is done using a Hidden Markov Model (HMM) or a majority voter (MV), as implemented in skweak (Lison et al., 2021).

### 5.3 Weakly supervised models

Handcrafted functions remain hampered by their limited coverage and lack of robustness to noise. *Weak supervision* can partially alleviate those limitations. Weak supervision operates by defining labeling functions and applying those on large amounts of unlabeled data to create a silver corpus, which is in turn employed to train a machine learning model for the task. We use the same 39 labelling functions as above and apply them to a set of 2,169 additional dialectal tweets collected similarly to the training data. Note that this data was not annotated by hand and serves mainly as a way to increase the size of the silver data with the hope of increasing recall. The outputs of those

---

[3]We rely here on the Norsk Ordbank for both Bokmål (https://www.nb.no/sprakbanken/ressurskatalog/oai-nb-no-sbr-5/) and Nynorsk (https://www.nb.no/sprakbanken/ressurskatalog/oai-nb-no-sbr-41/) and extract all inflected forms from those.

| Model | Dev | | Test | |
|---|---|---|---|---|
| 'Vowel shift' | 3.7 | | 4.4 | |
| Labeling functions (MV-aggregated) | 15.6 | | 16.4 | |
| NB-BERT fine-tuned on HMM-aggregated weak labels | 14.1 | ± 0.3 | 21.2 | ± 0.7 |
| NB-BERT fine-tuned on MV-aggregated weak labels | 29.7 | ± 0.6 | 33.3 | ± 0.7 |
| SVM + NB-BERT embeddings (gold labels) | 45.5 | | 47.7 | |
| BiLSTM fine-tuned on train (gold labels) | 38.5 | ± 3.4 | 45.5 | ± 0.0 |
| NorBERT fine-tuned on train (gold labels) | 42.0 | ± 6.0 | 52.9 | ± 1.3 |
| NB-BERT fine-tuned on train (gold labels) | 54.9 | ± 0.8 | 58.4 | ± 0.4 |

Table 2: Micro $F_1$ on dev and test for the vowel shift baseline, handcrafted labelling functions, weakly supervised models aggregated with either Hidden Markov Models (HMM) or majority voting (MV), and supervised models (BiLSTM, NorBERT, NB-BERT) trained on gold labels from the training set. The results for neural models are shown as the average and standard deviation of five runs with different random seeds.

functions are then aggregated using either HMMs or majority voting. After aggregation, we train an NB-BERT (Kummervold et al., 2021) model on this silver data using the same procedure as the supervised models described in the next section.

### 5.4 Supervised models

We test one context-free model and three sequence labeling models which take context into account: a bidirectional LSTM and two Norwegian pretrained language models. Those models are all fine-tuned on the gold labels of the training set.

The context-free model is an linear SVM trained using the embeddings from the NB-BERT model (see below). Specifically, we create vector representations for each word in the training data by passing the words individually to the embedding layer of NB-BERT. For words that are split into several subcomponents due to the byte pair tokenization, we take the average representation of these embeddings. Finally, we train a linear SVM classifier[4] and fine tune the C parameter on the dev set. This model therefore uses the same representation strategy as the stronger NB-BERT model, but uses these without contextualization and has significantly fewer trainable parameters.

The BiLSTM is a two layer Bidirectional LSTM (Schuster and Paliwal, 1997) with 100-dimensional pre-trained embeddings,[5] and a hidden layer size of 256. The embeddings were trained on the Norwegian Newspaper corpus, the Norwegian Web as corpus (NoWaC) (Guevara, 2010), and NBDigital corpus (books from the national library of Norway),

using fastText Skipgram (Bojanowski et al., 2017), and with a vocabulary size of 4,428,648 tokens. We train the BiLSTM model for a maximum of 50 epochs with a patience of 3 using Adam (Kingma and Ba, 2014) with default parameters.

The transformer models include NorBERT (Kutuzov et al., 2021) and NB-BERT (Kummervold et al., 2021). NorBERT is a BERT (Devlin et al., 2019) model trained from scratch, including the subword tokenizer, on the Norwegian Newspaper corpus combined with Wikipedia dumps for Bokmål and Nynorsk, for a total of nearly 2 billion tokens. The NB-BERT model is a multilingual BERT base model further trained on the Norwegian Colossal Corpus.[6] The latter is therefore less adapted to Norwegian vocabulary, but has been exposed to a larger volume and variety of Norwegian texts, including dialectal context.

As commonly done, we add a classification head to the transformer models and rely on the Huggingface library (Wolf et al., 2020) for the implementation. To deal with subword tokens, we assign the token label only to the first subword and mask the others. We use a learning rate of 2e-5, a weight decay of 0.01, and a batch size of 16 with Adam W (Loshchilov and Hutter, 2019). We train the models for 20 epochs, updating both pretrained weights and classification heads, and do not tune any parameters on the development set.

### 6 Results

Table 2 shows the micro-average $F_1$ scores obtained by all approaches on the test set.

The majority label baseline ('vowel shift')

---

[4]https://scikit-learn.org/stable/modules/generated/sklearn.svm.LinearSVC.html

[5]Model 81 downloaded from the NLPL word embedding repository http://vectors.nlpl.eu/repository/

[6]https://github.com/NbAiLab/notram/blob/master/guides/corpus_description.md

achieves a low $F_1$ score of 4.4. While the hand-crafted functions obtain slightly higher $F_1$ scores than these baselines, the scores demonstrate that the proposed task is challenging and that simple rule-based approaches are insufficient.

All supervised models perform better than the weak supervision models, with the BiLSTM achieving 45.5 $F_1$, the SVM 47.7, NorBERT 52.9, and NB-BERT 58.9. In general, the results of the SVM follow a high-precision low-recall pattern (*e.g.*, h-v: precision 90/recall 42, interjection: 100/8.3, palatalization: 100/13.3) displaying this model's inability to generalize to new examples, while the neural models tend to generalize better. The good performance of NB-BERT follows previous trends for classification of tweets (Barnes et al., 2021). Those results differs from Demszky et al. (2021), who found that the weak supervision provided by several hundred minimal pairs was often enough to outperform supervised approaches. This discrepancy may be due to differences in the training set size or the increased difficulty of labeling the tokens rather than the full utterance.

| Label | Precision | Recall | $F_1$ |
|---|---|---|---|
| copula | **94.5** | **94.8** | **94.7** |
| pron. subj. | **82.9** | **74.3** | **78.4** |
| pm deletion | **72.4** | **79.9** | **76.0** |
| pron. obj. | **88.2** | 63.8 | **74.0** |
| h-v | 67.4 | 69.0 | 68.2 |
| functional | **71.2** | 63.9 | 67.3 |
| voicing | **73.7** | 58.3 | 65.1 |
| apocope | **75.5** | 53.6 | 62.7 |
| nom. decl. | 66.0 | 55.6 | 60.4 |
| dem. pro. | 60.0 | 60.0 | 60.0 |
| contraction | **77.1** | 45.8 | 57.4 |
| vowel shift | 58.4 | 55.3 | 56.8 |
| phon. spelling | 40.7 | 36.5 | 38.5 |
| shortening | 41.3 | 35.2 | 38.0 |
| adj. decl. | 36.8 | 28.0 | 31.8 |
| palatalization | **75.0** | 20.0 | 31.6 |
| interjection | 30.0 | 25.0 | 27.3 |
| conjugation | 24.3 | 15.8 | 19.1 |
| marked | 6.7 | 8.0 | 7.3 |
| lexical | 50.0 | 3.0 | 5.7 |
| gender | 0.0 | 0.0 | 0.0 |

Table 3: Precision, recall, and $F_1$ scores of NB-BERT.

# 7 Error Analysis

We provide here an error analysis of the results from the best performing model, namely NB-BERT. Table 3 shows the per-label precision, recall, and $F_1$ scores of the NB-BERT model. We highlight scores > 70 in blue and scores < 50 in red. The model performs well on *copula*, *pronouns* (subject and object), and *present marker deletion*. It performs poorly on *phonemic spelling*, *shortening*, *adjectival declension*, *interjection*, *conjugation*, *marked*, *lexical*, and *gender*. There is a statistically significant correlation between frequency in the training corpus and $F_1$ (Spearman's $\rho = 0.65$, $p = 0.001$), although there are outliers such as *vowel shift*. This may be due to the range of heterogeneous contexts in which *vowel shift* can occur. Other labels such as *functional* or *h-v* are more difficult than expected, likely due to the number of possible forms.

It is clear from the confusion matrix in Figure 4 that the model confuses most labels with the label 'Ø'. The other label that is regularly over-predicted is 'vowel shift', which suggests that frequency plays a strong role in prediction. When it comes to multiple labels, the performance of NB-BERT can be characterized as high-precision and low-recall, with only 30% of the test tokens with multiple labels being predicted as such by the model, with a micro $F_1$ of 88.1.

To establish the importance of context, we compare the performance of the SVM and NB-BERT models on context-free labels (h-v, functional, vowel shift, voicing, palatalization, shortening, interjection, nominal declension, conjugation, marked, lexical) and context-sensitive labels (phonemic spelling, contraction, pronoun subject, pronoun object, present marker deletion, apocope, adjectival declension, demonstrative pronoun, copula, gender). We compare these two groups by taking the average difference between the F1 scores for each label. For the context-free labels, there is an average 14.0 percentage points difference between the two models, while for the context-sensitive labels, this difference is 24.9. This implies that including context via contextual embeddings is especially important for the context-sensitive labels.

# 8 Conclusion and future work

In this paper, we have presented a new dataset for token-level dialect feature prediction, composed of Norwegian tweets classified as dialectal, which we annotated for 21 dialectal features achieving good

| true \ pred | Ø | adj-decl | apocope | conj | contr | copula | dem-pron | funct | gender | h-v | interj | lexical | marked | nom-decl | palat | phonemic | pm-delet | pron-obj | pron-subj | shortening | voicing | vowel-shift |
|---|---|---|---|---|---|---|---|---|---|---|---|---|---|---|---|---|---|---|---|---|---|---|
| Ø | 10391 | 9 | 5 | 12 | 2 | 3 | 1 | 25 | | 6 | 5 | 1 | 31 | 42 | | 41 | 32 | 3 | 26 | 3 | 2 | 70 |
| adj-decl | 13 | 7 | 1 | 1 | | | | | | | | | | | | | | | 1 | | | 2 |
| apocope | 19 | | 37 | 2 | 2 | | | | | | | | | 2 | | 1 | | | 3 | | | 4 |
| conj | 25 | | 4 | 9 | | 1 | | | | | | | | | | 5 | | | | | | 3 |
| contr | 12 | | | | 27 | 1 | | 1 | | 1 | 1 | | 1 | | | 4 | | | 6 | 2 | | 5 |
| copula | 5 | | 1 | 1 | 1 | 257 | | | | | | | | | | 2 | | | 3 | 1 | | 4 |
| dem-pron | 3 | | | | | | 6 | | | | | | | | | | | | 1 | | | |
| funct | 28 | | | 3 | 2 | 3 | | 205 | 8 | 1 | | | 2 | | | 8 | 7 | 2 | 7 | 10 | 1 | 52 |
| gender | 6 | | | | | | | | | | | | | | | | | | | | | 2 |
| h-v | 4 | | | | | | | 5 | | 29 | | | 1 | | | 1 | | | | | | 2 |
| interj | 9 | | | | | | | | | 3 | | | | | | | | | | | | |
| lexical | 16 | | | 1 | | | | 1 | | | 1 | | 2 | 2 | | 1 | | | | 1 | | 16 |
| marked | 26 | 1 | | 1 | | | | 5 | | | | | 4 | 12 | | | | | | 1 | | 14 |
| nom-decl | 75 | | | | | 2 | | | | | | | | 140 | | 8 | | | 8 | | | 22 |
| palat | 3 | | | | 1 | 2 | | | | | | | 1 | 3 | | | | | 1 | | | 5 |
| phonemic | 29 | 1 | | 1 | | | | 9 | | | | | 5 | 6 | | 46 | 2 | | 1 | 6 | | 18 |
| pm-delet | 25 | | 1 | 2 | | 1 | | 1 | | | | | | | | 1 | 147 | | | | | 6 |
| pron-obj | 16 | | | | 1 | 8 | | | | | | | | | | 1 | | 67 | 8 | | 1 | 6 |
| pron-subj | 64 | | | 2 | 6 | 2 | | 3 | | | 1 | | | | | | | 4 | 243 | | | |
| shortening | 11 | | | 2 | | 6 | | | | 1 | | | 2 | 2 | | 1 | | | | 19 | 1 | 4 |
| voicing | 3 | | 1 | 1 | | | | | | | | | | | | | | | | | 14 | 8 |
| vowel-shift | 98 | 2 | | 7 | | 1 | | 39 | | 2 | | | 24 | 14 | 2 | 18 | 16 | | 3 | 8 | | 296 |

Figure 4: Confusion matrix of the NB-BERT model. 'Ø' represents predicting no label. The y-axis represents the true labels.

inter-annotator agreement. This dataset was employed in a set of labelling experiments including rule-based approaches, weakly supervised, and supervised neural models. The experimental results corroborate the difficulty of the task, with micro $F_1$ scores ranging from 16.4 for handcrafted functions to 58.9 for the best supervised model.

This work provides a basis for future research on dialectal features. Specifically, we plan to explore the distribution of these dialectal features in different online communities using the learned models. The data can also help multi-task learning of text normalization models, as identifying tokens to be normalized should lead to improvements.

Another promising direction is to predict regional dialects based on the token-level features. As dialectal traits are correlated with certain regions, it may be possible to create hierarchical representations of dialects on different levels of granularity. The guidelines, models, and annotations will be made publicly available.[7]

As for potential risks, the dataset was compiled from social media posts. Therefore, complying with the GDPR regulations, authors of these posts must have the right to be forgotten if they wish to remove previous posts. We will therefore only release the annotations with the original tweets upon request. In this way, if they have been deleted, they will also not be recuperated for our dataset.

## Acknowledgements

This work has been partially supported by the Teksthub initiative at the University of Oslo, MediaFutures, the HiTZ center and the Basque Government (Research group funding IT-1805-22).

We also acknowledge the funding from the following projects: DeepKnowledge (PID2021-127777OB-C21) project funded by MCIN/AEI/10.13039/501100011033 and by FEDER Una manera de hacer Europa.

Parts of this work was supported by industry partners and the Research Council of Norway with funding to MediaFutures, project number 309339.

Finally we want to thank the two annotators Alexandra Wittemann and Marie Emerentze Fleisje for their annotation efforts.

---

[7] https://github.com/jerbarnes/nordial

## Appendix A – Limitations

Our motivation for this project was to take a first step towards fine-grained dialectal feature detection. However, there are several limitations with the current annotation process and modeling approaches presented in this paper.

Firstly, although the idea of identifying dialectal features in Twitter data is rather general, the guidelines and dataset provided with this paper are specific to Norwegian. While we hope that these resources are helpful to other language variations, adapting this to another situation would require a non-trivial amount of work and money. The creation of this dataset required 7000 euro.

The annotation procedure focused on token-level labels. Dialectal features that arise from the absence of a given token (*e.g.* subject dropping, as in Example 10) or that cannot be marked at the token-level (*e.g.* non-V2 word order in interrogative sentences as in Example 11) are therefore not explicitly annotated in this dataset.

(10) *Spent   på   det   ...*
     Exited  on   it    ...
     '(I am) excited about it'

(11) *Ka    du    sier?*
     What  you   say?
     'What are you saying?'

## Appendix B – Co-occurence of annotated labels

Figure 5 shows the co-occurrence of the 21 labels at token-level. From the figure, it is clear that most labels do not co-occur or do so rarely. The labels that co-occur the most frequently are vowel shift and functional (366), vowel shift and present marker deletion (121), functional and contraction (104), phonemic and functional (65) and pronoun subject and contraction (57). Vowel shift, besides being the most common label, is also the label that co-occurs the most with other labels.

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

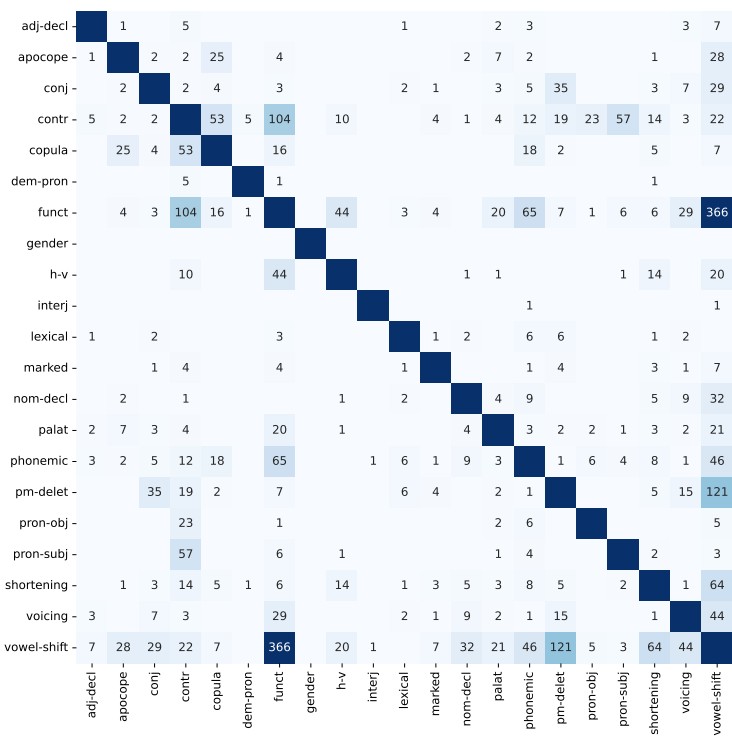

Figure 5: Co-occurrence of annotated labels.

Jacob Devlin, Ming-Wei Chang, Kenton Lee, and Kristina Toutanova. 2019. BERT: Pre-training of deep bidirectional transformers for language understanding. In *Proceedings of the 2019 Conference of the North American Chapter of the Association for Computational Linguistics: Human Language Technologies, Volume 1 (Long and Short Papers)*, pages 4171–4186, Minneapolis, Minnesota. Association for Computational Linguistics.

Gabriel Doyle. 2014. Mapping dialectal variation by querying social media. In *Proceedings of the 14th Conference of the European Chapter of the Association for Computational Linguistics*, pages 98–106, Gothenburg, Sweden. Association for Computational Linguistics.

Jacob Eisenstein. 2013. Phonological factors in social media writing. In *Proceedings of the Workshop on Language Analysis in Social Media*, pages 11–19, Atlanta, Georgia. Association for Computational Linguistics.

Jacob Eisenstein. 2015. Systematic patterning in phonologically-motivated orthographic variation. *Journal of Sociolinguistics*, 19:161–188.

Jacob Eisenstein, Brendan O'Connor, Noah A. Smith, and Eric P. Xing. 2010. A latent variable model for geographic lexical variation. In *Proceedings of the 2010 Conference on Empirical Methods in Natural Language Processing*, pages 1277–1287, Cambridge, MA. Association for Computational Linguistics.

Mihaela Gamăn, Sebastian Cojocariu, and Radu Tudor Ionescu. 2021. UnibucKernel: Geolocating Swiss German jodels using ensemble learning. In *Proceedings of the Eighth Workshop on NLP for Similar Languages, Varieties and Dialects*, pages 84–95, Kiyv, Ukraine. Association for Computational Linguistics.

Sidney Greenbaum and Gerald Nelson. 1996. The international corpus of English (ICE) project. *World Englishes*, 15(1):3–15.

Emiliano Raul Guevara. 2010. NoWaC: a large web-based corpus for Norwegian. In *Proceedings of the NAACL HLT 2010 Sixth Web as Corpus Workshop*, pages 1–7, NAACL-HLT, Los Angeles. Association for Computational Linguistics.

René Haas and Leon Derczynski. 2021. Discriminating between similar nordic languages. In *Proceedings of the Eighth Workshop on NLP for Similar Languages, Varieties and Dialects*, pages 67–75, Kiyv, Ukraine. Association for Computational Linguistics.

Dirk Hovy and Christoph Purschke. 2018. Capturing regional variation with distributed place representations and geographic retrofitting. In *Proceedings of the 2018 Conference on Empirical Methods in Natural Language Processing*, pages 4383–4394, Brussels, Belgium. Association for Computational Linguistics.

Naman Jain and Riyaz Ahmad Bhat. 2014. Language identification in code-switching scenario. In *Proceedings of the First Workshop on Computational Approaches to Code Switching*, pages 87–93, Doha, Qatar. Association for Computational Linguistics.

Tommi Jauhiainen, Heidi Jauhiainen, and Krister Lindén. 2021. Naive Bayes-based experiments in Romanian dialect identification. In *Proceedings of the Eighth Workshop on NLP for Similar Languages, Varieties and Dialects*, pages 76–83, Kiyv, Ukraine. Association for Computational Linguistics.

Taylor Jones. 2015. Toward a Description of African American Vernacular English Dialect Regions Using "Black Twitter". *American Speech*, 90:403–440.

Diederik Kingma and Jimmy Ba. 2014. Adam: A method for stochastic optimization. In *Proceedings of the 3rd International Conference on Learning Representations (ICLR)*.

Klaus Krippendorff. 1980. *Content Analysis: An Introduction to Its Methodology*. Sage Publications, Beverly Hills, CA, USA.

Per E Kummervold, Javier De la Rosa, Freddy Wetjen, and Svein Arne Brygfjeld. 2021. Operationalizing a national digital library: The case for a Norwegian transformer model. In *Proceedings of the 23rd Nordic Conference on Computational Linguistics (NoDaLiDa)*, pages 20–29, Reykjavik, Iceland (Online). Linköping University Electronic Press, Sweden.

Andrey Kutuzov, Jeremy Barnes, Erik Velldal, Lilja Øvrelid, and Stephan Oepen. 2021. Large-scale contextualised language modelling for Norwegian. In *Proceedings of the 23rd Nordic Conference on Computational Linguistics (NoDaLiDa)*, pages 30–40, Reykjavik, Iceland (Online). Linköping University Electronic Press, Sweden.

William Labov. 2006. *The Social Stratification of English in New York City*, 2 edition. Cambridge University Press.

Pierre Lison, Jeremy Barnes, and Aliaksandr Hubin. 2021. skweak: Weak supervision made easy for NLP. In *Proceedings of the 59th Annual Meeting of the Association for Computational Linguistics and the 11th International Joint Conference on Natural Language Processing: System Demonstrations*, pages 337–346, Online. Association for Computational Linguistics.

Ilya Loshchilov and Frank Hutter. 2019. Decoupled weight decay regularization. In *ICLR*.

Yann Mathet, Antoine Widlöcher, and Jean-Philippe Métivier. 2015. The unified and holistic method gamma for inter-annotator agreement measure and alignment. *Computational Linguistics*, 41(3):437–479.

Preslav Nakov, Marcos Zampieri, Nikola Ljubešić, Jörg Tiedemann, Shevin Malmasi, and Ahmed Ali, editors. 2017. *Proceedings of the Fourth Workshop on NLP for Similar Languages, Varieties and Dialects (VarDial)*. Association for Computational Linguistics, Valencia, Spain.

Preslav Nakov, Marcos Zampieri, Liling Tan, Nikola Ljubešić, Jörg Tiedemann, and Shervin Malmasi, editors. 2016. *Proceedings of the Third Workshop on NLP for Similar Languages, Varieties and Dialects (VarDial3)*. The COLING 2016 Organizing Committee, Osaka, Japan.

John Nerbonne. 2009. Data-driven dialectology. *Language and Linguistics Compass*, 3(1):175–198.

Dong Nguyen and Jacob Eisenstein. 2017. A kernel independence test for geographical language variation. *Computational Linguistics*, 43:567–592.

Younes Samih, Suraj Maharjan, Mohammed Attia, Laura Kallmeyer, and Thamar Solorio. 2016. Multilingual code-switching identification via LSTM recurrent neural networks. In *Proceedings of the Second Workshop on Computational Approaches to Code Switching*, pages 50–59, Austin, Texas. Association for Computational Linguistics.

Helge Sandøy. 2000. Utviklingslinjer i moderne norske dialektar. *Folkemålsstudier*, 1(39):345–384.

Mike Schuster and Kuldip Paliwal. 1997. Bidirectional recurrent neural networks. *IEEE Transactions on Signal Processing*, 45:2673 – 2681.

Thamar Solorio and Yang Liu. 2008. Learning to predict code-switching points. In *Proceedings of the 2008 Conference on Empirical Methods in Natural Language Processing*, pages 973–981, Honolulu, Hawaii. Association for Computational Linguistics.

Pontus Stenetorp, Sampo Pyysalo, Goran Topić, Tomoko Ohta, Sophia Ananiadou, and Jun'ichi Tsujii. 2012. BRAT: A Web-based Tool for NLP-assisted Text Annotation. In *Proceedings of the Demonstrations at the 13th Conference of the European Chapter of the Association for Computational Linguistics*, pages 102–107, Avignon, France.

Rachael Tatman. 2016. "I'm a spawts guay": Comparing the Use of Sociophonetic Variables in Speech and Twitter. *University of Pennsylvania Working Papers in Linguistics*, 22.

Hadrien Titeux and Rachid Riad. 2021. pygamma-agreement: Gamma $\gamma$ measure for inter/intra-annotator agreement in Python. *Journal of Open Source Software*, 6(62):2989.

Kjell Venås and Martin Skjekkeland. 2022. dialekter i noreg i store norske leksikon på snl.no. https://snl.no/dialekter_i_Noreg. Accessed: 2020-09-30.

Thomas Wolf, Lysandre Debut, Victor Sanh, Julien Chaumond, Clement Delangue, Anthony Moi, Pierric Cistac, Tim Rault, Remi Louf, Morgan Funtowicz, Joe Davison, Sam Shleifer, Patrick von Platen,

Clara Ma, Yacine Jernite, Julien Plu, Canwen Xu, Teven Le Scao, Sylvain Gugger, Mariama Drame, Quentin Lhoest, and Alexander Rush. 2020. Transformers: State-of-the-art natural language processing. In *Proceedings of the 2020 Conference on Empirical Methods in Natural Language Processing: System Demonstrations*, pages 38–45, Online. Association for Computational Linguistics.

Marcos Zampieri, Shervin Malmasi, Nikola Ljubešić, Preslav Nakov, Ahmed Ali, Jörg Tiedemann, Yves Scherrer, and Noëmi Aepli. 2017. Findings of the VarDial evaluation campaign 2017. In *Proceedings of the Fourth Workshop on NLP for Similar Languages, Varieties and Dialects (VarDial)*, pages 1–15, Valencia, Spain. Association for Computational Linguistics.

Marcos Zampieri, Shervin Malmasi, Preslav Nakov, Ahmed Ali, Suwon Shon, James Glass, Yves Scherrer, Tanja Samardžić, Nikola Ljubešić, Jörg Tiedemann, Chris van der Lee, Stefan Grondelaers, Nelleke Oostdijk, Dirk Speelman, Antal van den Bosch, Ritesh Kumar, Bornini Lahiri, and Mayank Jain. 2018. Language identification and morphosyntactic tagging: The second VarDial evaluation campaign. In *Proceedings of the Fifth Workshop on NLP for Similar Languages, Varieties and Dialects (VarDial 2018)*, pages 1–17, Santa Fe, New Mexico, USA. Association for Computational Linguistics.
