# OpenReview forum: "Identifying Token-Level Dialectal Features in Social Media"
_NoDaLiDa/2023/Conference — NoDaLiDa 2023_

### Official Review · Reviewer_8m5i · 2023-02-28
**A corpus of Norwegian tweets annotated on token level with dialect features and a series of experiments on the associated task of dialect feature identification**

**Rating:** 8
**Confidence:** 5

**Review:**

This paper presents a corpus of Norwegian tweets which are annotated with dialect features on token level. Several experiments are conducted on the associated task of dialect feature identification (i.e. not dialect identification), and the task is found to be challenging because of its multi-label nature and sparsity of some of the labels. To my knowledge, this is the first paper proposing this particular task, which is a welcome addition to the better-known but simpler task of dialect identification. While the 21 identified dialect features are specific to Norwegian, the annotation protocol and the experimental setup could be easily applied to datasets from other languages featuring internal variation.

Pros:
- The paper proposes a potentially attractive new task in the area of linguistic variation.
- The annotation process is well explained and documented with helpful examples.
- The experiments show that many dialectal features are challenging to identify.

Cons:
- The dataset does not seem to be annotated with dialect areas or with geographical metadata (e.g. geotags or metadata in the user's profile). While this information is not necessary for the proposed task, it could make the dataset much more valuable for related tasks such as dialect identification or geolocation prediction. It could also help us assess the sometimes questioned reliability of such metadata. I think this is a missed opportunity.
- Some aspects of the experimental setup are not clear to me, e.g. concerning the aggregation of the handcrafted functions and the multilabel conversion - see below.

Questions:
- L545: I would assume that models suffer equally from having 159 output classes, most of which are only seen in 1 or 2 examples (if I interpret Figure 4 correctly). I wonder if other approaches wouldn't make the learning easier, e.g. randomly selecting one of the labels for each token, or only use merged labels if they have >n occurrences in the dataset?
- L585: I don't understand why you need this sophisticated aggregation procedure. Why not just take the union of all proposed labels, and output 'O' if that set is empty? Moreover, if some of the functions themselves are context-sensitive already (e.g. L565), what would be the benefit of an HMM?
- Table 2: Is the label 'O' included in the micro-average computation or not? If it is, then wouldn't the most common label baseline have to refer to 'O'? (Maybe I just got confused because O is included in Table 3...)

Typos/minor remarks:
- L152: The VarDial series has continued after 2018 - maybe include "e.g." in the parentheses
- L155: Mihaela et al. > Gaman et al. (names are switched in the bib entry)
- L240: Do you have a specific source for this example?
- §4.1: It wasn't immediately clear to me what exactly the label is - as far as I understand the bolded paragraph headers correspond to the labels? Maybe you could add a fully annotated example sentence in the appendix showing the annotation and label format?
- L349: Why 'but'? I don't understand why there is a contrast between the two sentences...
- L370: labels > label
- L382: it seems to me that one l of 'still' is bold-faced
- §4.1: Are the features ordered in a particular way? It could make sense to (roughly) order them according to their frequency (cf. Figure 2)
- L467-473 (possibly also Table 1): Maybe this could be simplified to just include 3 percentages: no label, 1 label, >1 labels.
- L532-538: 2 sentences with the same content
- Table 3: The numbers are not well aligned. Maybe you can put the neither-blue-nor-red numbers into a rectangle with white background?
- Figure 4: It could be helpful to have the absolute numbers of single-label annotations on the diagonal to get a better intuition of the proportions.

**Paper Type:**

Long paper

---

### Official Review · Reviewer_Xu5c · 2023-03-07
**Paper on annotating and identifying dialectal variation in written text on social media**

**Rating:** 7
**Confidence:** 5

**Review:**

The paper addresses the issue of identifying dialectal variation in text written on social media. This is an important contribution as it presents an interesting opportunity for future studies on large-scale digital dialectology.  Overall, the paper well written and has a clear motivation and delimitation of the topic, which is really appreciated, but does lack clarity and methodological considerations on some topics.

*Pros*
- Important contributions presenting future directions for studies within digital dialectology
- Well-grounded in linguistics
- Solid annotation guidelines and experimental setup with bootstrapping and error analysis

*Cons*
- Missing motivation/theoretical grounding for utilizing contextual word embeddings for the task of predicting dialectal features.


The evaluation will refer to the three main contributions of the paper, namely, (1) The introduction of the novel task of the prediction of dialectal features of tokens, (2) establishment of a new annotated corpora of tweets in Norwegian, and (3) an evaluation of the learnability of dialectal features.

Common to all three contributions are that they are original and well-grounded in linguistics and are significant contributions to the area of digital dialectology as they present a possible framework for performing such studies large scale.

The methodological considerations behind the second contribution are thorough and well-documented, explaining how the annotation guidelines were established and including metrics on inter-annotator agreement.

The first and third contributions, however, lacks clarity and methodological grounding:

*Section 4*
In general, it is unclear to the reader what the labels of dialectal features look like. While the authors provide a list of examples of dialectal variation in Norwegian, the task and annotation is not generally introduced. For example, the section starts a bit abruptly with an example from English, but it is not clear to the reader what this example is supposed to demonstrate. While I presume, it is to demonstrate the task and the annotations it includes, the example does not highlight this, as the actual annotation is not included. I would have liked to have seen this.

*Section 5*
- The section lacks a more formal definition of the learning task, e.g., the input is a list of words/tokens W = {w_1, w_2 .., w_t}, and the output is a list of labels L={O, Copula, …, O}. As the task is presented as a multilabel classification task, this should be specified by the formal definition.
- L546 What is the label ‘O’ – I am not sure that this is generally introduces, although I suspect that it is the normative/null/Ø/0 label.
- L546 What is your merging strategy?

*Section 5.3+5.4*
What is your reasoning behind using word embeddings for predicting dialectal features? In particular:

- For features that are e.g., phonological grounded, how would the contextual embedding of a word encode information relevant for the prediction of that feature? Is it because the surrounding words can be seen as predictive? Or do you presume that this information is captured in the static embedding layer itself? And for the latter: Why would you presume that for morphological/phonological variation?
- How do you expect tokenization to affect your results? How many of the dialectal words are tokenized into different sub-tokens? And doing a qualitative analysis of these sub-tokens: Do they make sense for the task? Especially, noting that you are only using the first sub-token for prediction.

*Other comments*
- L253 How do you determine dialectal impact? "We focus on the dialectal impact a word has, i.e. whether the annotator can determine that the word falls outside of the norms in such a way as to identify the speaker as a dialect user."
- Figure 3: Include axis labels (True/Predicted)
- It would be interesting to see the correlation between f1 and number of types for each category, as this could be an indicator on the difficulty of the task.



**Paper Type:**

Long paper

---

### Official Review · Reviewer_mc7F · 2023-03-09
**Corpus and experiments for developing token level dialectal feature label identification**

**Rating:** 6
**Confidence:** 5

**Review:**

Pros:
- Dialectal features annotated on the word level would be useful in many tasks
- The annotation guidelines and resulting annotations appear quite good

Cons:
- NLP research on code-switching is ignored in the previous work section
- The experimental setup is not clearly described

Section 2: You ignore code-switching research in NLP, which is language identification on word level. There are other word-level language identification articles out there as well, but most of the research has been done labeled as code-switching. You should take a look at this: https://aclanthology.org/W14-3907/ and some of the more than 200 articles that refer to it. Especially the standard Arabic vs. dialectal Arabic.

155: Mihaela is the first name of Mihaela Gaman. This should be "Gaman et al.".

178: "presents" -> "present".

209: Why did you collect further 3,000 tweets?

229: "were classified" or "were classified as" something?

551: Isn't the majority label no label at all? Or are you labeling only those words that were labeled by the annotators? You are not describing the experimental setup clearly enough.

621-624: References or links needed here for the corpora.

625: References or links needed here for fastText.

789: What is label 'O'?

Figure 3: You should declare which axis has the true labels.

Figure 3: How is this a confusion matrix of errors. Aren't the correct predictions here as well? Like the 296 vowel-hift ones? From these numbers I'm starting to doubt what was the evaluation setting in the first place. Are you only testing on the tokens that were identified as dialectal by the annotators? What would be this kind of classifier ever used? At least on line 844 you mention detecting the dialectal tokens themselves, which is a worthy goal, but the Figure 3 numbers do not add up to this task.

855: Very goog to point these out, but do GDPR regulations apply to Norway? Anyway, if you're in EU, the data-mining directive gives you the right to retain the data and non-publicly distribute it for research purposes. Retaining the author-ID's with the data might be unnecessary. I would move the last paragraph from Conclusions to Appendix A.

**Paper Type:**

Long paper

---

### Decision · Program_Chairs · 2023-03-17

Accept